# MECHANISMS OF SKILL TRANSFER FROM PRETRAINING TO TARGET TASKS IN RECURRENT NEURAL NETWORKS

## ABSTRACT

Pretraining on simpler tasks can often improve learning outcomes on a more difficult target task. Nonetheless, what makes for a good pretraining curriculum and the mechanisms of positive transfer across tasks remain poorly understood. Here we use RNNs trained on fixed length temporal integration to compare curricula with varying degrees of effectiveness. We show that pretraining on simpler versions of the target task is less effective than curricula which take advantage of the target task's compositional structure and train sub-skills needed for solving it. By exploiting the highly structured solution of our target task, we can mechanistically explain improvements in speed and quality of learning in terms of the slow features of the RNN dynamics that the curriculum helps build, and the reuse and adaptation of those slow features during target training. Our results argue that pretraining on tasks that individually hone sub-skills required for the target are particularly beneficial, as they build a scaffolding on which additional dynamical systems structures can be compositionally expanded to achieve the final function. Thus, our results document a novel mechanism for repurposing dynamical systems features in support of cognitive flexibility.

## 1 INTRODUCTION

In biological systems, learning of a new skill always happens on top of a structured body of previous knowledge. Moreover, when training animals in new experimental tasks, this preexisting knowledge is purposefully supplemented by first training simpler relevant tasks. Such behavioral shaping often proves critical for being able to learn a desired behavior within a limited time. It is also an increasingly common approach to training recurrent neural network (RNN) models on complex tasks (Krueger & Dayan, 2009). Outside biology, using pretraining or some form of curriculum learning is a common strategy for improving the quality of training in many machine learning tasks (Soviany et al., 2022; Hacohen & Weinshall, 2019; Narvekar & Stone, 2018). What makes for a good pretraining task and the mechanisms by which knowledge is reused across tasks remains poorly understood.

Curriculum learning (CL)—the strategy of organizing training examples from simple to complex—has long been recognized as an effective approach to accelerate learning (Bengio et al., 2009). Traditional curriculum methods focus on gradually increasing task difficulty through shorter sequences (Bengio et al., 2015; Chan et al., 2015), reduced data complexity, or automated difficulty progression (Graves et al., 2017; Haviv et al., 2019). The mechanics of why CL might help are not fully understood but it is typically explained through the lens of the loss landscapes that different tasks induce: simpler versions of the same task have easier to navigate loss surfaces bringing the model parameters at good initial conditions for the more difficult to optimize loss surface of the target task. Although this perspective has been influential in understanding why simpler tasks aid training, open questions remain about what the process of transfer looks like from the perspective of the representations learned during pretraining relative to those used in the final solution.

The reuse of pretrained representations has recently been the focus of study in multi-task RNNs. Such models can learn reusable dynamical motifs and rule structures (Yang et al., 2019; Driscoll et al., 2024), where complex behaviors emerge through flexible combination of these dynamical sys-

tems computational elements. The geometric organization of these learned representations depends critically on network architecture and task structure: networks may either reuse shared subspaces across tasks or develop orthogonal representations (Turner & Barak, 2023; Vafidis et al., 2025). Moreover, the structure of the initial conditions can determine the speed and nature of the learning process (Liu et al., 2024). Despite much progress, cognitive flexibility remains largely studied in scenarios where the recurrent dynamics (and associated dynamical systems computation motifs) are structured but fixed. A new task then learns to repurpose these elements through learnable inputs and outputs (Driscoll et al., 2024). It remains unclear how new dynamical systems motifs can be learned on top of an already structured dynamical systems and what kind of dynamics repurposing is possible in a sequential task learning context.

The ability to combine learned primitives to solve novel problems is tightly related to task compositionality (Lake & Baroni, 2018; Zhou et al., 2024; Park et al., 2024), which provides a principled approach for constructing useful curricula. Evidence from animal learning (Boyd-Meredith et al., 2025), human behavioral studies (Szabó & Fiser, 2025), and computational models (Hocker et al., 2025; Mark et al., 2020) demonstrates that pretraining on tasks that target specific sub-computations or relational structures substantially improves subsequent learning of complex tasks. How this happens at the level of learned neural representations is only partially understood (Hocker et al., 2025). When are the already existing primitives enough for new task adaptation vs. when *de novo* learning of additional structure is needed is not always clear, although both strategies have been documented biologically (Chang et al., 2024; Yang et al., 2021; Gastrock et al., 2024).

Here we aim to understand positive benefits of RNN pretraining in terms of the network's dynamical systems features and how they change over learning. We do so by exploiting the very particular dynamical systems structure of a new variant of temporal integration to mechanistically investigate how pretraining on simpler tasks shapes the internal representations of RNNs at the level of slow dynamical systems features that support task relevant computations. We identify a collection of different curricula that all prove beneficial in terms of speeding up learning in the target task. Different curricula have different mechanistic ways of achieving knowledge transfer. Compositional curricula yield the strongest benefits for target task training, by ensuring low rank effective changes in the network dynamics. These correspond to a dynamic scaffold of useful function that then gets adapted through the addition of further complementary dynamic modes during in task training. Moreover, different sequential curricula which exploit compositional structure can lead to qualitatively different dynamics repurpose strategies, from lazy reuse of exiting primitives, to rich reorganization of the circuit dynamics as a whole.

## 2 METHODS

### 2.1 PROBING EFFECTS OF PRETRAINING WITH A FIXED-LENGTH INTEGRATION TASK

We use a standard discrete time RNN, with network states $\boldsymbol{h}_t \in \mathbb{R}^N$ evolving as:

$$\boldsymbol{h}_t = (1 - \alpha)\boldsymbol{h}_{t-1} + \alpha f(\boldsymbol{W}_{rec}\boldsymbol{h}_{t-1} + \boldsymbol{W}_{in}\boldsymbol{x}_t + \boldsymbol{b}), \tag{1}$$

where $\boldsymbol{x}_t \in \mathbb{R}^2$ is the input, $\alpha$ is the leak rate, tanh nonlinearity $f(\cdot)$, and trainable parameters $\boldsymbol{W}_{rec}, \boldsymbol{W}_{in}, \boldsymbol{b}$. The output $\boldsymbol{y}_t \in \mathbb{R}^1$ is a linear readout $\boldsymbol{y}_t = \boldsymbol{W}_{out}\boldsymbol{h}_t$. The recurrent weight matrix $W_{rec}$ was initialized either deterministically to a rescaled version of the identity matrix, while $W_{in}$ and $W_{out}$ were initialized using Xavier uniform initialization. We also tested Xavier uniform initialization for $W_{rec}$ and found qualitatively consistent results across all curriculum conditions, but chose diagonal initialization as it removes one sort of variability from the process allowing us to focus on the effects of pretraining in changing those initial conditions. For all experiments, $N = 100$, with a leak $\alpha = 0.9$.

Our target task family is a variant of evidence integration, where independent Gaussian noise inputs need to be summed-up over a time period $[1, T]$ to generate the output (Fig. 1A), $y_t^{\text{int}} = \sum_{i=1}^{t} x_i$ where $x_t^{\text{stim}} \sim \mathcal{N}(\mu, \sigma_{\text{stim}}^2)$. In each trial the mean input $\mu$ is randomly chosen as $\mu = \pm\mu_0$ with equal probability. An additional input channel signals beginning of a new trial with a impulse at $t = 1$, which provides a mechanism for a dynamic reset of the network state at the beginning of a new trial. (see Appendix A for training details).

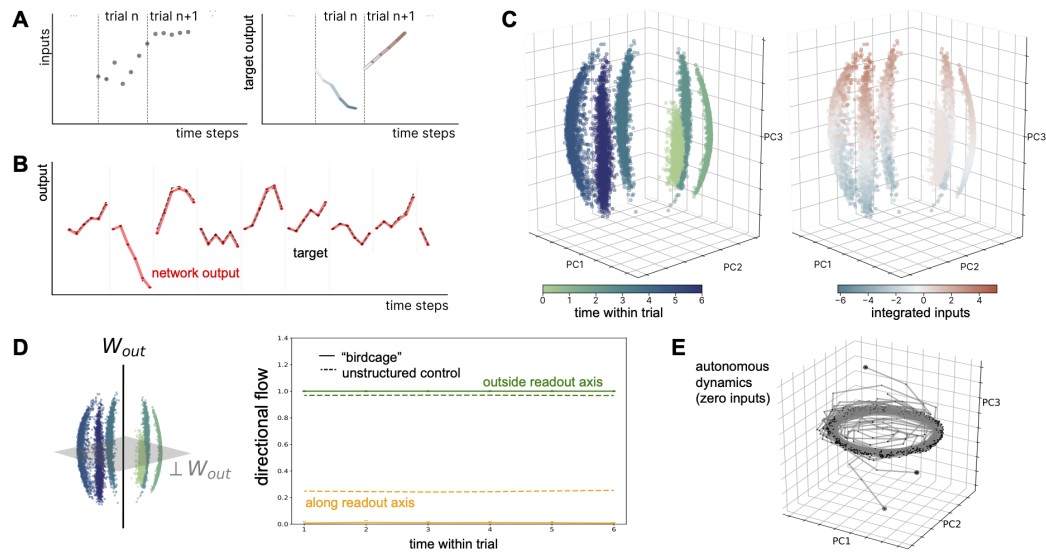

Figure 1: An integration task with structured dynamics. (A) Example input-output trials for the task: inputs are i.i.d. gaussian samples, while the output needs to report their sum over the course of the trial. (B) Example network performance after training. (C) PCA projection of the network states, color coded by either time within trial (left), or target output (right). (D) Projection of the average network flow along the vertical (readout) axis and in the orthogonal horizontal plane as a function of time within trial. Control network does not have birdcage structure. (E) Example trajectories of autonomous dynamics starting at typical initial trial start states; larger dots show trajectories for perturbed initial conditions.

Two features make it different from the standard version of this task: 1) the output needs to be reported throughout the trial as opposed to reporting the final sign of the sum at the end (Sussillo & Barak, 2013; Bredenberg et al., 2024), and 2) all trials have equal duration, with trial length, $T$, controlling overall task difficulty. While this task is too simple to strictly require pretraining, we will show that this twist on the standard formulation can still capture some interesting knowledge transfer scenarios. Moreover, the simplicity of the setup allows the effects to be understood through the lens of repurposing and adapting preexisting dynamical system structure.

## 2.2 A HIGHLY STRUCTURED DYNAMICAL SYSTEMS SOLUTION

While solo training of the integration task leads to good target output reconstruction (Fig. 1B), the dynamical system structure of the solution is somewhat variable (for networks trained on trial length 6 integration tasks, 11 out of 25 developed non-birdcage solutions). One particularly interesting strategy that the trained networks seem to develop in solving the task takes the form of a "birdcage" in the network activity space (Fig. 1C). Each vertical "bar" marks one time point within the trial (left), with the position of activity along the bar directly mapping into the integrated output (right).

This is interesting since it seems more structured than it needs to be: it is not immediately clear why representing the passage of time would be useful for robustly performing the task; the fixed trial duration is a robust statistical regularity in the training data so this cyclic nature can be perhaps exploited for more effective task resetting.[1] This additional structure does come with additional complications: for this geometry to be able to actually perform the task, the intermediate sum of inputs up to the current step needs to be maintained and updated as the dynamics move from bar to bar. If traditional evidence integration achieves the computation with a single (functional) line attractor (Bredenberg et al., 2024), this variant of the task needs a one-dimensional slow dynamic mode along each of the bars.

---

[1]Note that in a solution relying on one line attractor, resetting from below or from above zero would need inputs with opposite sign, which is not possible with a single linear reset signal.

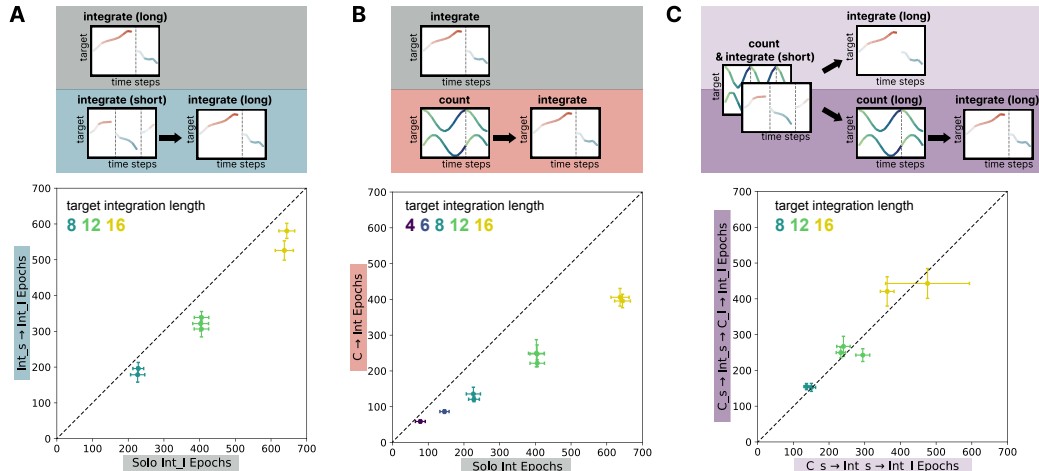

Figure 2: Learning speedups when pretraining with different curricula. Top: cartoon illustration of the curriculum comparison; Bottom: numerical results averaged across 25 seeds. (A) Traditional pretraining with shorter delays for the same task. (B) Pretraining with the counting time sub-task, with length matched to integration target. (C) Mixed curricula combining short counting and integration with an optional target matched long counting. Learning speed measured by no. of epochs required to reach threshold convergence on the target task, starting from random initial conditions. Colors indicate target integration length. Multiple data points of the same color reflect different pretraining configurations (e.g., varying pretraining task lengths) leading to the same target task.

To make these intuitions mathematically precise, we evaluated the flow of the autonomous network dynamics (no input) starting from natural initial conditions, at different time points within the trial (corresponding to different bars). Specifically, the flow defined as vector $\Delta \boldsymbol{h}_t = \boldsymbol{h}_{t+1} - \boldsymbol{h}_t$, which measures the direction and magnitude of state changes in the network at any time point, or alternatively by position in state space. We partition this total flow into the vertical component along the $W_{out}$ axis and the remainder, which includes flow along the "equator" of the birdcage structure and whatever residual flow happens in higher dimensions of activity (Fig. 1D). We find virtually zero autonomous flow along the vertical axis, which means that that axis of the dynamics behaves functionally like a set of line attractors where shifts in the output are fully driven by new input coming in. In contrast, the flow within the horizontal plane is non-negligible and consistent in magnitude across time, reflecting a steady transition from bar to bar. Tracing zero input trajectories along the "equator" in response to perturbed unusual initial conditions suggests that the dynamics are attractive from outside of the birdcage structure, making the horizontal plane flow functionally a limit cycle (Fig. 1E). These signatures are reduced or missing in control networks that still perform the task well but have no clear structure in their first 3 PCs, possibly a reflection of a functionally equivalent but higher dimensional (and potentially less robust) solution (Fig. 1D, dashed lines). Overall, our variant of integration shows highly structured and directly measurable dynamical systems features in its solution, whose emergence we can hope to trace back through the learning process when using different pretraining curricula.

## 3 ACCELERATING LEARNING BY SCAFFOLDING NEURAL DYNAMICS

### 3.1 SEVERAL CURRICULA SPEED UP TRAINING IN OUR TASK

What would a pretraining curriculum look like for our simple integration task? The most obvious answer goes back to the essence of the CL idea (Bengio et al., 2009) which is to start small, with a simpler version of the same task. This "short integration" pretraining starts by training for a small $T_1$ up to reasonable performance before switching to the longer $T_2$ target task (Fig. 2A). An alternative idea motivated by behavioral shaping (Hocker et al., 2025; Krueger & Dayan, 2009) is to break out the final solution into its compositional sub-elements and design a pretraining task intended to hone

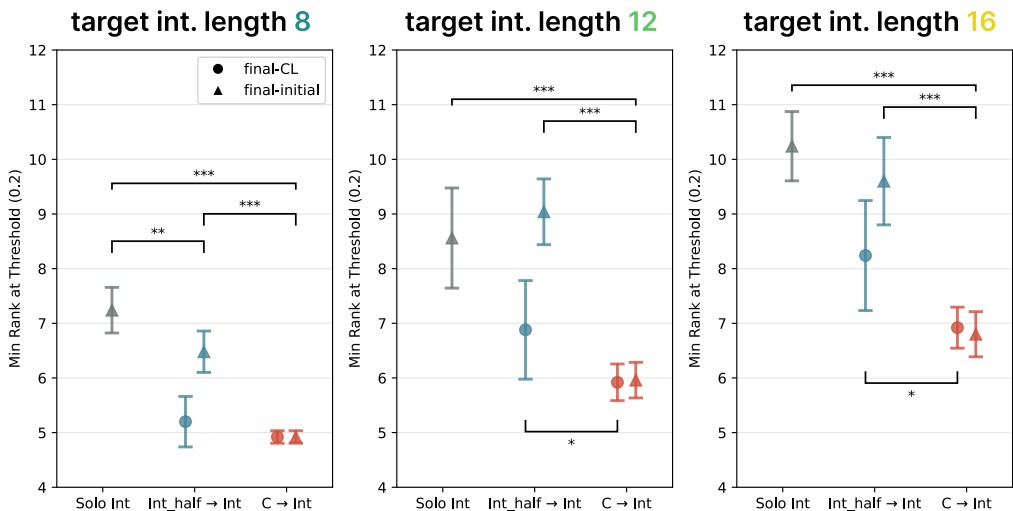

Figure 3: Dimensionality of functional changes in dynamics, as measured by the minimum rank approximation of the learning-induced recurrent changes needed to preserve final task function. Parameter changes measured relative to random initial conditions (triangles) for full learning effects, and relative to pretrained state (circle) as measure of within-task learning. Left to right: different target integration lengths.

in those skills individually. Since good solutions in our task often take advantage of representing time within a trial, we decided to pretrain networks using a simple counting task, in which the network needs to report the phase of an oscillation with period $T$ in two output channels, $y_t^{\cos} = \cos(2\pi(t-1)/T)$ and $y_t^{\sin} = \sin(2\pi(t-1)/T)$ (Fig. 2B); for simplicity, the trial length is matched with the target. Finally, we also consider combinations of the two tasks as more complex mixed curricula which also introduce mismatched lengths between counting and target integration(Fig. 2C.

Perhaps unexpectedly given the simplicity of the setup, we find that all curricula considered improve over solo target task training in terms of speed of learning, as measured by the number of epochs required to reach criterion performance on the target task starting from random initial conditions. This is true across a range of target task difficulty levels, $T$, but the magnitude of the improvements depends on the pretraining procedure. Benefits are modest for short integration but much more substantial when using counting as part of the pretraining, especially at long trial lengths. These benefits saturate with more complex curricula, where the addition of long counting after short counting and short integration does not seem to further improve speed of convergence (Fig. 2C).

### 3.2 DIFFERENT CURRICULA INDUCE CHANGES WITH DIFFERENT EFFECTIVE RANKS

It is well understood that learning simple tasks induces low rank changes to the RNN recurrent weights (Schuessler et al., 2020). Moreover, it was recently shown that the rank of the initial conditions for the weights can change the nature of learning for a given task, interpolating between rich and lazy learning (Liu et al., 2024). We wondered thus if it would be possible to understand differences in the magnitude of speed-ups of different curricula in terms of the rank of the changes they induce to the network dynamics over learning.

How low rank are changes introduced by different learning curricula? To answer this, we turned to a metric of the minimum *necessary rank* of weight changes induced during learning needed to support final task structure (Schuessler et al., 2020). This concretely replaces the full change in recurrent parameters $\Delta W$ with increasingly low rank approximations $\Delta W_k = \sum_{i=1}^{k} \sigma_i \mathbf{v}_i \mathbf{v}_i^T$, where $\sigma_i$ are the eigenvalues and $\mathbf{v}_i$ are the corresponding eigenvectors of $\Delta W$.[2] It then asks what is the

___
[2]The input weights $W_{in}$ and output weights $W_{out}$ were kept fixed at their final trained values.

lowest rank approximation such that the corresponding recurrent weights still preserve good task performance, within a pre-specified tolerance.

We considered two variants of this metric: the traditional version which measures the parameter change induced by the full training process, from random initial condition to final task convergence (Fig. 3, triangles); this should describe the dimensionality of the dynamic modes used to solve the final task. A second version of this analysis measures the necessary rank of parameter changes specifically when training on the target task (Fig. 3, circles). This provides an intuitive notion of "richness"/"laziness" in that if pretraining has already developed some of the dynamical systems structures needed for the target task, then within task learning can proceed quickly and with very low rank changes to the network dynamics.

Across task difficulties, we find a very systematic difference in the minimum necessary rank for the full course of learning between count-based pretraining versus alternatives. This implies that the counting curriculum results in systematically more compact dynamics for the solutions that it finds for the target task. In contrast, the dimensionality of short integration curricula has much more similar ranks to solo training. The degree of reorganization during target task training was also different across curricula, with integration requiring very substantial reorganization of the dynamic modes (of rank on pair with solo training) whereas the target specific adaptation was very low rank in the counting curricula. While the results presented use a ratio of 2 between the short and long intervals, similar phenomena can be observed for other choices of short length and other curricula (Suppl. Fig. 7). Overall, pretraining using counting seems to lead to more compact solutions and comparatively low rank dynamic changes during target task training. This is likely a reflection of representational refinement rather than *de novo* learning.

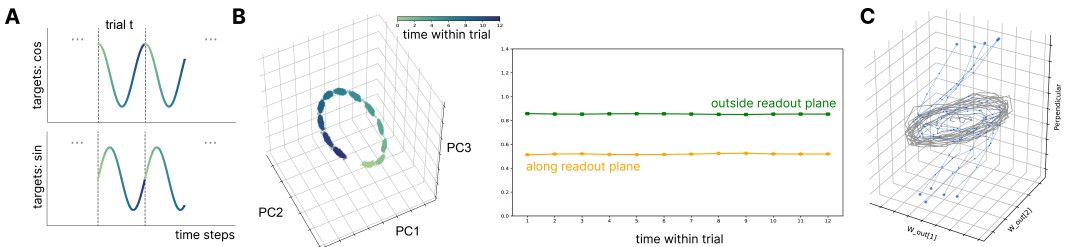

Figure 4: Different curricula lead to different dynamical systems structure. (A) The target output for the counting task involves a 2-d periodic output; inputs are the same as for integration but completely irrelevant for counting. (B) Low dimensional projection of population activity after training, colored by oscillation phase (left) and associated dynamical systems characterization (right) where the directional flow is estimated relative to the 2-d count readout axes. (C) Trajectories driven by autonomous dynamics for typical initial conditions (gray) and in response to vertical perturbations (blue). The z is defines as the axis of largest variance orthogonal to the readout plane.

### 3.3 DYNAMICAL SCAFFOLDS AND COMPOSITIONAL GENERALIZATION

To understand the mechanism by which counting pretraining leads to compact and faster-to-learn representations, we turned once more to the geometry of network states and their dynamics (Figure 4). Since the network receives the same kinds of inputs during pretraining as in the target task (even if they act as a nuisance from the perspective of the pretraining task), we would expect that the dynamics at the end of counting pretraining would not only exhibit periodicity but that they would try to clamp the integration input channel, or at least place those inputs into the null space of the relevant network responses. Indeed, the representation learned via counting shares the periodic nature of the temporal representation, but without the vertical bars seen in the final task solution. Investigating the network flow along the "horizontal" plane of the task relevant outputs, we find a consistent flow akin to the functional limit cycle seen for integration. Unlike integration, the orthogonal axis shows substantial autonomous flow along the "vertical" axis. Moreover, given that the network dynamics seem to compensate for perturbations along that dimension, it is likely that the flow outside of the output plane reflects attractor forces that pull the dynamics onto the limit cycle generating the counting outputs (Figure 4).

This seems very counterintuitive: we took one idiosyncratic property of the target solution, i.e. exploiting fixed trial length, and used it to build a pretraining task that not only reinforces that aspect of the dynamics but does so at the expense of penalizing dynamical structure that would be desirable for the end goal. And yet, learning the target from the resulting starting point is still much faster than any alternative. A way to reconcile these observations is to think of them in terms of compositionality of the dynamic modes: if the counting pretraining builds one dimension of the dynamics in the form of the effective limit cycle element, the target tasks can use very low-dimensional perturbation of those dynamics to add the one extra slow dimension needed for the integration (the vertical span of the birdcage). Thus, the primary mechanism for skill transfer in this setting is the preservation of existing dynamic modes paired –which provides a dynamic scaffolds of sorts– with a low rank expansion of the dynamics to account for added new functionality. Moreover, curricula involving counting tasks yield consistent dynamical structures across different initializations, as evidenced by the low variance in phase trajectories (Suppl. Fig. 8), suggesting that the temporal scaffolding provided by counting leads to more robust solutions. More generally, the structured curricula reduce across-seed variability in terms of the effects of task training (in terms of ranks, representational structure and any other metrics we have measured) providing a more narrow but speedy path towards a good final solution.

### 3.4 ADAPTING EXISTING STRUCTURE VS. BUILDING NEW ONE

Up to this point, we have focused our mechanistic understanding of CL on pairs of tasks with a shared trial length. What happens when the length of trials in pretraining is shorter than that of the final integration task (as is always the case for short integration curricula)?

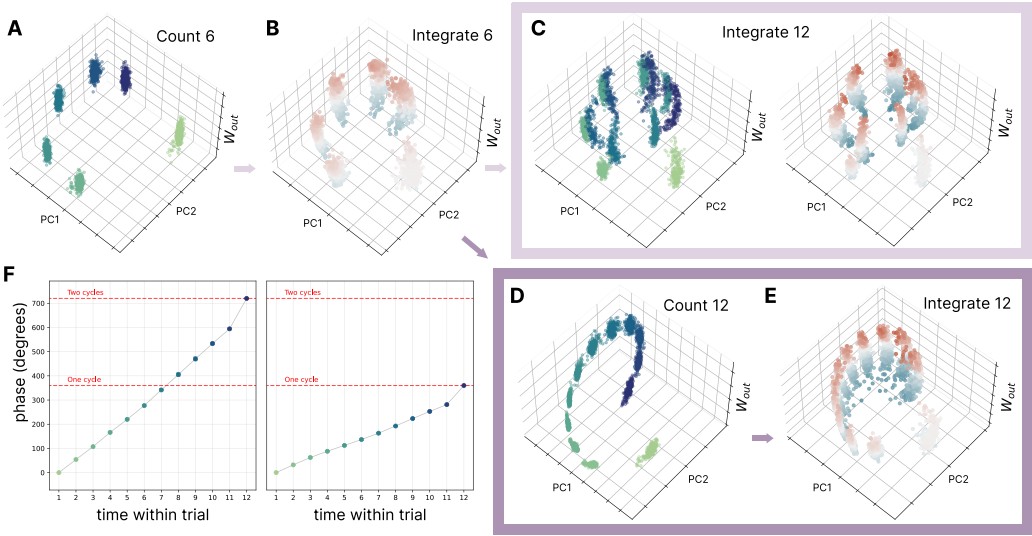

Figure 5: Dynamic feature reuse in multi-task curricula. (A-E) Network activity structure across multiple stages of pretraining for integrate-12 target task. (F) Phase of the population activity as a function of time within trial for the Count6→Integrate6 (left) vs. Count6 → Integrate6→ Count12 (right) pretraining; mean and sem estimated across 25 seeds.

A particularly illustrative example is the version where the trial length doubles in the target task relative to pretraining, $T_2 = 2T_1$ (Fig. 5). Concretely, we start with training counting for length 6, which builds a circular geometry but no encoding of integrated outputs, then expand the corresponding birdcage vertical bars via training integration with the same length. At this point of the process, we either jump straight into the target task integration length 12, or include additional pretraining for counting with length 12.

The two curricula sequences yield systematically different mechanisms of task adaptation. In the first scenario, the dynamics straight out reuse the existing dynamical systems structure where activity circles the 6 bar birdcage twice. The concurrent presence of the start input, together with the network

being in the period-end state, is enough to reset the dynamics for a new trial. In contrast, training with the long counting task reorganizes the representation to a length 12 limit cycle which then expands an additional dimension of the integrated output, as (see also Suppl. Fig. 7B). This is a statistically robust result across seeds (Fig. 5F), showing that different pretraining procedures induce strong inductive biases in terms of the nature of the final solution that the RNN learns for a given target task (see also Suppl. Fig. 8).

The first type of adaptation is fundamentally lazy, by exploiting the representational task alignment of the curriculum to effectively learn nothing new. The second causes compact, low dimensional but very structured reorganization of the representation in the service of a new task (in this case Integrate6 → Count12). The dynamic reuse demonstrated above suggests that networks can leverage existing dynamical system structure (as well as low rank) to accelerate learning on related tasks. The lack of additional speedups with the longest curriculum (including counting to 12) relative to its direct integration counterpart (Fig. 2C) can be understood as a tradeoff between reusing already existing structure directly which is a little slower to train vs. pretraining further to make in task training ever so slightly faster. By the time networks have completed short counting and short integration, they have already established the essential dynamical scaffolds needed for the target task: a functional limit cycle for temporal representation and the capacity for integration along orthogonal dimensions; further reorganization provides minimal learning efficiency benefit, but at the cost of additional training time. Nonetheless, representational differences between them remain relevant in terms of the priming of the network for future learning. In particular, we expect that the long curriculum will lead to networks that are faster to generalize to even longer temporal integration windows.

### 3.5 SHARED REPRESENTATIONAL SUBSPACES ACROSS TASKS

While we have substantial evidence that dynamical systems features built during pretraining get reshaped and reused for learning in the final task, whether the topological reuse of structure comes with systematic geometric changes is not clear. To investigate this in more detail, we analyzed the similarity of the network's representational geometry at different stages of the curriculum (Fig. 6). Our analysis compared the structure of a single network after Count6 pretraining to its final structure after subsequently learning the Integrate6 task (Fig. 6A, blue). As a null model for the magnitude of these effects, we compared the final states of two networks that were independently trained on the full curriculum from different random seeds (Fig. 6A, red). First, we evaluated the similarity of the overall representation subspaces generated by the hidden state activity. We used three complementary metrics for this: 1) the alignment of principal component axes, 2) the degree of subspace overlap, and 3) Centered Kernal Alignment (CKA) (Kornblith et al., 2019). The different metrics all paint a coherent picture: they show significantly more aligned geometry between the network's representation at the end of pretraining and the final solution relative to control (Fig. 6B). Furthermore, to examine the proximity of individual learned trajectories, we calculated the Euclidean distance between network's evolution of states in several ways. Specifically, we computed the distance between corresponding hidden states for each of the six time steps within a trial, averaged these six values to obtain an overall trajectory distance, and additionally calculated the distance between the mean hidden state vectors of each model (Fig. 6C). As for all other quantifications, we find that network trajectories are geometrically preserved over the final target learning process.

## 4 DISCUSSION

In this work, we investigated the mechanisms by which curriculum learning shapes the internal dynamics of RNNs in the service of speeding up learning. We showed that although many curricula can induce some degree of speedup relative to solo task training for our simple temporal integration, what kind of dynamical system structure they build and how that gets reused by the target task can vary. The most important and counterintuitive result is that pretraining on counting –which aims to build one of the task-required dynamical modes at the cost of another– yields the strongest benefit. Mechanistically, this provides a dynamic scaffold (in this case a limit cycle that keeps track of time within trial) which gets combined with a new line-attractor extended dimension to implement the target function. This provides not only a possible explanation for the empirical benefits of compositional curricula (Hocker et al., 2025), but also a counterpart for the RNN simplicity bi-

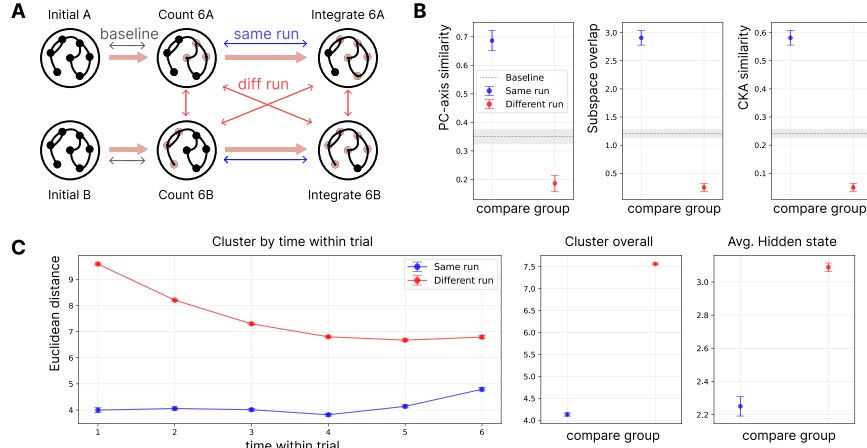

Figure 6: Networks share representational geometry across curriculum stages. (A) Experimental design: Networks initialized from different random seeds undergo curriculum learning, Count6 →Integrate6. Comparisons are made within individual curriculum sequences (same run, blue) versus across different random initializations (different run, red). Gray indicates initial to count-trained comparison. (B) Subspace similarity analysis across three measures-PC axis similarity, subspace overlap, and CKA similarity. (C) Euclidean distance analysis between hidden states under zero input noise conditions.

ases documented previously (Turner & Barak, 2023), but through the lens of compositionality of dynamics: New dynamic modes build up on top of existing ones to achieve complex function.

This feature distinguishes our approach from recent RNN models of cognitive flexibility which demonstrate effective dynamics reuse through explicit context signals or rule inputs (Yang et al., 2019; Driscoll et al., 2024). In our case, the dynamical primitives remain plastic and can be constantly be reshaped by new experience. The new learning happens in very compact spaces (parameter changes being low rank) which may have interesting implications for continual learning in terms of the ability of the systems to learn multiple unrelated tasks without interference. Future work will need explore this in more detail.

From the perspective of the target task, pretraining can be thought of as a mechanism for favorable parameter initialization. This perspective aligns our findings with recent results on the effects of initial low-rank connectivity on learning outcomes (Liu et al., 2024). This connection between curriculum design and initial condition engineering suggests potentially broader conceptual relationships with modern mathematical attempts at understanding RNN learning (Proca et al., 2025).

The neural tangent kernel framework (Jacot et al., 2018) reveals a dichotomy between lazy and rich learning regimes—minimal versus substantial feature reorganization. Critically, initial weight structure determines which regime dominates (Liu et al., 2024), with consequences for solution efficiency and generalization. While the distinction between rich and lazy is not always clear in our setup, we were able to identify several qualitatively different scenarios: 1) direct reuse of an existing dynamical system feature (count and integrate joint curriculum), 2) reshaping of dynamic modes on top of existing structure (Count$T$ to Integrate$T$) and 3) de novo formation of more complex structure (e.g. long curriculum including Count12). These argue for new metrics of laziness in RNN training, focused on the persistence of topological features of the dynamics across the learning process, perhaps akin to those that have been recently developed for studying metadynamics in single tasks (Marschall & Savin, 2023).

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

## A  EXTENDED METHODS

**Loss function and optimization.** All networks were trained using Mean Squared Error (MSE) loss between the network output and target function. We used the Adam optimizer with a learning rate of 0.0001 and default hyperparameters ($\beta_1 = 0.9$, $\beta_2 = 0.999$).

**Training protocol.** Training was conducted in batches of 32 trials. Each epoch consisted of 512 batches. Rather than training for a fixed number of epochs, convergence for each task or curriculum stage was determined adaptively: a network was considered converged when its test set MSE remained below 0.2 for 10 consecutive epochs. This threshold was chosen to ensure robust task performance while allowing comparison across different curricula.

**Data generation.** Each batch contained 200 time steps total, yielding $200/T$ trials per batch (e.g., 33 trials for $T = 6$, 16 trials for $T = 12$). Stimulus inputs were drawn from $\mathcal{N}(\mu_0, \sigma_{stim}^2)$ with $\mu_0 = 0.3$ and $\sigma_{stim} = 0.6$, where the sign of $\mu_0$ was randomly chosen per trial with equal probability.

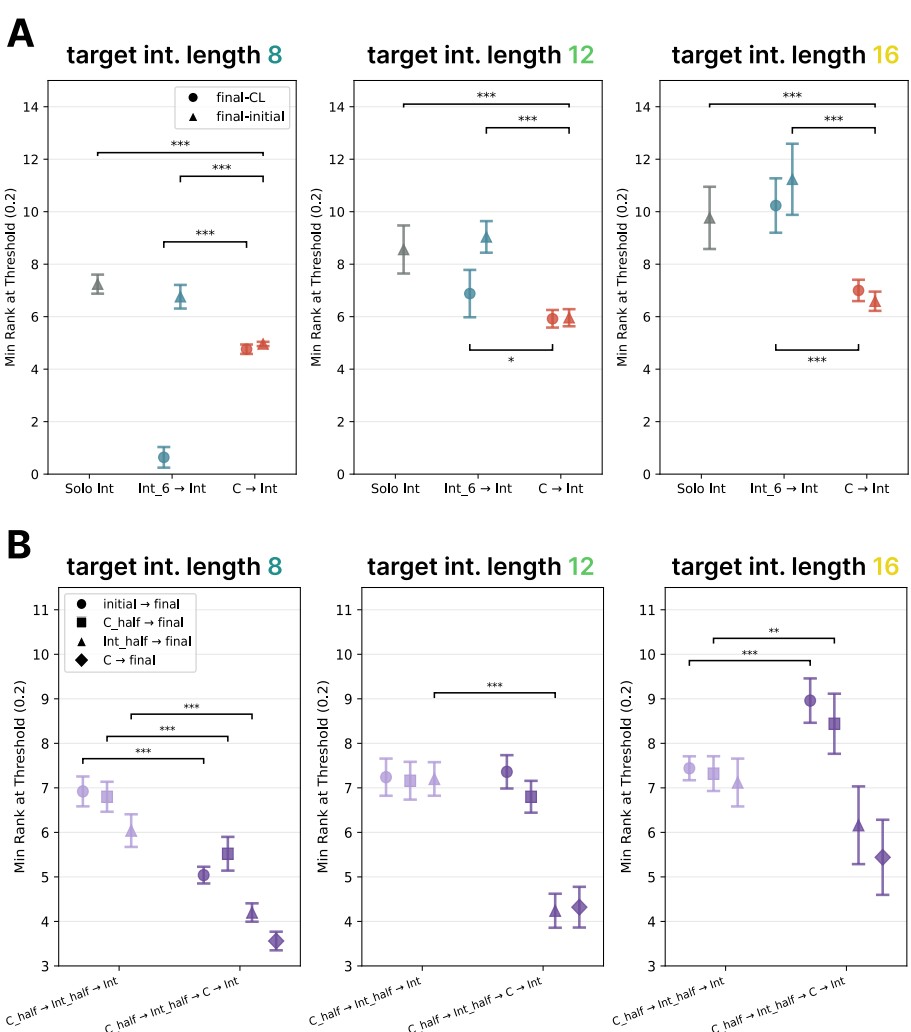

Figure 7: Effective rank for different curricula. (A) Minimal rank for different training curricula; conventions as for Fig.3, but for different relationships between pretraining and target trial length. (B) Same as A but covering the different stages of the complex pretraining curricula described in Fig. 2C.

## B  BENEFITS OF LEARNING SPEED COME FROM REUSE OF DYNAMICAL SYSTEMS STRUCTURE, NOT JUST LOW DIMENSIONALITY

Given the observations that pretraining shapes the minimal rank of the computation, and with it the dimensionality of the recurrent dynamics, as well as constructing slow dynamical systems features that seem to get reused in the target task, it is reasonable to wonder what exactly drives the increase in speed? Is it the low dimensionality of the dynamics, or the actual dynamical systems structures built within that low dimensional manifold?

To investigate this distinction, we used a two dimensional version of the Flip-flop task as pretraining (Fig.9). Training on this task results in a well-documented solution with four geometrically symmetric fixed points forming a square (Maheswaranathan et al., 2019). Input dependent transitions allow the system to move between these points implementing a two bit form of item working memory. The solution for this task shares many features with our Count4 pretraining: both have a planar geometry and four slow features within it, with the key distinction that the nature of the slow features

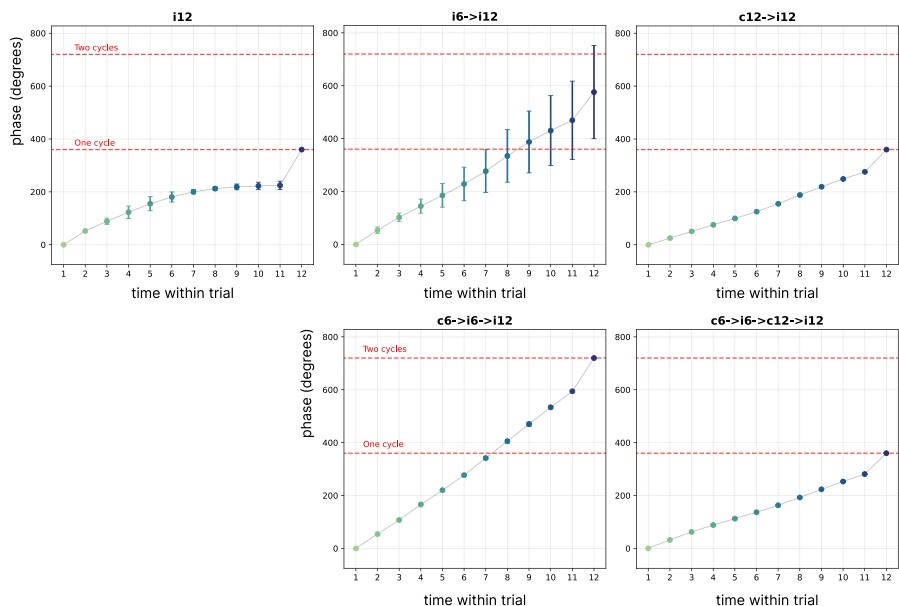

Figure 8: Geometry of the final network trajectories for different curricula. Phase of the population activity as a function of time within trial across different curriculum sequences. Top row: Integrate12 (solo), Integrate6→Integrate12, Count12→Integrate12. Bottom row: Integrate6→Count6→Integrate12, Integrate6→Count6→Count12→Integrate12.

is different. Counting dynamics have a consistent rotational flow between the slow points, whereas the Flip-flop task do not have such flow. Importantly, the benefits of pretraining are only seen for counting (regardless of the number of time steps, see also Fig. 2B). Despite the many similarities, it takes a larger number of epochs to train the integration target task ((Maheswaranathan et al., 2019) a, left) which reflects the need for more dramatic changes in topology needed to construct the final solution. Starting from the flip-flop solution as initial conditions still provides an improvement over starting from scratch ((Maheswaranathan et al., 2019) a, gray) which proves that the corse geometry is part of the benefit, but the flip-flop is generally slower to train, bringing the total training time up by a large margin. Mechanistically, the integration solution after flip-flop training does not seem to reuse the full set of fixed points, but rather concentrates the integration computation around one of them ((Maheswaranathan et al., 2019) b). Overall, these results suggest that knowledge transfer in this task primarily relies on the reuse and adjustment of the limit cycle motif from pretraining to the final birdcage solution.

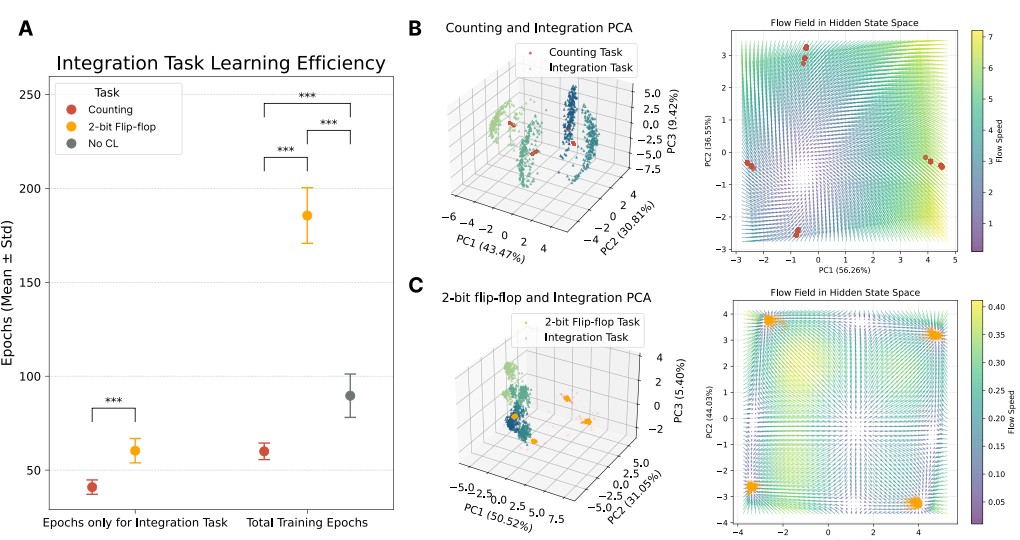

Figure 9: Fixed points from 2-bit flip-flop pretraining fail to provide dynamical scaffolding for curriculum transfer. (A) Learning efficiency comparison across three conditions: Counting→Integration (red), 2-bit Flip-flop→Integration (orange), and No CL (gray). Left: epochs for integration task only (post-pretraining); Right: total training epochs including pretraining. Error bars show mean ± std across 25 seeds. Counting pretraining significantly reduces both integration-specific and total training time, while 2-bit flip-flop pretraining even increases total training time due to the cost of learning an incompatible dynamical structure. (B) Counting task case. Left: combined PCA of both counting task (red) and integration task (green) hidden states, showing overlapping structure. Right: flow field for recurrent activity in the space of the first two PCs after counting pretraining, showing limit cycle dynamics. (C) 2-bit flip-flop task case. Left: combined PCA of both 2-bit flip-flop task (orange) and integration task (green) hidden states, showing distinct structure. Right: flow field for recurrent activity in the space of the first two PCs after flip-flop pretraining, revealing convergence to four stationary attractors.

