# OpenReview forum: "Mechanisms of skill transfer from pretraining to target task in recurrent neural networks"
_ICLR.cc/2026/Conference — Submitted to ICLR 2026_

### Official Review · Reviewer_MAk9 · 2025-10-15

**Soundness:** 2
**Presentation:** 2
**Contribution:** 1
**Rating:** 2
**Confidence:** 5

**Summary:**

This paper provides a mechanistic perspective on how pretraining shapes the dynamics of RNNs on a fixed-length temporal integration task. The authors argue that pretraining on compositional sub-skills (the counting task) builds a reusable dynamical scaffold (a limit cycle) upon which the target integration behavior can be built upon.

**Strengths:**

- Understanding mechanisms of transfer across tasks in RNNs is an important and timely question for computational neuroscience.
- The paper goes beyond studying the speed of learning and analyzes the dynamical features of the task to provide a mechanistic account of why pretraining on counting speeds up subsequent learning on the integration task.

**Weaknesses:**

1) Limited novelty and scope

Much of the arguments made by the authors has strong overlap with prior work (for example, Driscoll et al. (2023) on dynamical motif reuse in multitask RNNs; Turner & Barak (2023) on attractor reuse when the RNNs are trained sequentially on multiple tasks with overlapping task structure; Hocker et al. (2025) on compositional pretraining speeds up subsequent learning through reuse of dynamical features). The present contribution largely re-articulates these ideas on a single, relatively simple integration task with fixed trial lengths. The authors acknowledges that the task is “too simple to strictly require pretraining,” which undermines the general significance of the claimed mechanisms.

2) Unclear generalizability

The paper acknowledges that the task is “too simple to strictly require pretraining” (Page 3), raising questions about its broader relevance. The focus on low-rank changes and dynamic scaffolds might not translate to scenarios with higher-dimensional or less structured tasks, and this concern is not adequately addressed.

Other comments:
- On line 91, this is a discrete-time RNN.
- Can you also show the x-tick labels (the values of the input/output) in Fig. 1A–B?
- This paper would benefit a lot from clearer writing and better interpretation of the results. For example, in Fig. 3 the final-CL effective rank and the final–initial effective rank are always the same for the C→Int training regime. This is interesting because it implies that the necessary structure for performing the integration task may have been developed during the counting phase and then reconfigured to perform the full task, but the paper does not explicitly make this point.
- On line 215, the authors state that “good solutions in our task often take advantage of representing time within a trial.” However, it remains unclear why the networks develop a limit cycle to track the passage of time. Is this because multiple trials are concatenated with an impulse at the beginning of each trial to indicate a new trial, so that learning a time-tracking limit cycle helps the network reset its hidden state beyond using the impulse cue?

**Questions:**

See weaknesses

---

> ### Author Response · Authors · 2025-11-19
> **Reply to reviewer MAk9**
>
> Weaknesses
>
> At a distance what we do here might look similar to Driscoll but there is a critical distinction that makes our results fundamentally different: the recurrent weights are plastic throughout the training process. Driscoll looks at how plastic inputs and output can recycle function of a fixed bank of pre-trained dynamical systems features. We look at how existing dynamical systems feature influence the learning of new skills though both the reuse and the reshaping of existing functionality.  In our example, the reshaping happens without losing the original functionality: the limit cycle is still there so the network can still do counting, despite the addition of an extra slow axis orthogonal to the original structure. This points to a potential new axis of combinatorial expansion, where more slow dimensions can be added to existing dynamical systems motifs to expand the set of computations the circuit can perform.
>
> The use of the term ‘continuous time’ in describing the RNN refers to the autoregressive nature of the process for an intermediate value of alpha.
>
> We have added new results to expand on generalizability.
>
> Q1) the x-axis represents Time in time steps. Input values are sampled from a Gaussian distribution with a standard deviation of 0.6 and a trial mean of either +0.3 or -0.3. Output range given by the cumulative sum. These are arbitrary units so the scaling is not really important.
>
> Q2) We are sorry the reviewer found the text hard to follow but to be able to improve it some more concrete issues or suggestions would be needed.
>
> With regards to the mechanism comment: we do not think that all the necessary structure is learned in the counting task, only specifically the rotational flow aspect of the limit cycle. A new dimension of slow dynamics needs to be added on top during target training before the full functionality is achieved. If everything were already in place, there would not be the need for nontrivial amounts of in target task training to reach the final solution. Furthermore, there is nothing in the counting task that would encourage or require the additional slow axis (along the bars). If anything  those fluctuations are just noise during pretraining  and they should be quenched. It turns out that that axis lies in the null space of the readout and is not actively suppressed. Nonetheless, the extra slow modes needed for the bar are not there either. The partial reuse plus extra dynamical system dimensions is precisely what we argue in the text, see p6 l323- original; p7 l328- revised pdf
>
> Q3) The fixed trial length definitely helps drive the emergence of the limit cycle. The input start helps resetting but because of the fixed trial length it ends up being not strictly necessary. We were hoping that the explicit reset could be used instead of keeping track of time but there is a nontrivial geometric reason why that does not work: assume I use a traditional single line attractor solution for evidence integration plus one input pulse at the beginning of the trial as a means to reset. Positive mean trials will integrate rightwards from zero whereas negative mean trials will integrate values towards the left. A transient reset input would have to push the network state leftwards for the first type of trial but rightwards for the second but I have a single set of input weights to determine the directionality of its effects. I would need to have at least one additional state dimension to push in, orthogonal to the line attractor and anything along that off the line attractor axis to act as a sink towards the integration axis’ zero. Such a topology seems just as hard if not harder to implement. One could just assume a separate reset process that brute force resets the state to zero at the beginning of each trial but does not seem very biological. Our setup is also not the same as the evidence integration or decision making networks a la Wong & Wang 2006 where no input means that the network relaxes to a single stable zero fixed point etc Since the task requires the network to output the integration result at each step of the process and not just the sign of the final outcome at the end of the trial (unlike e.g. Bredenberg et al 2024 where the network learns to use a pair of stable fixed points with a flat energy barrier between them as a way to approximate a line attractor). Overall, it is counterintuitive how the exact instantiation of an evidence integration task variant maps into slow feature geometry of the solution. We see our version as a happy accident: the same way how the crystalline cube solution of the 3bit flip flop has enabled people to reason cohesively about what is the essence of a computation (Maheswaranathan et al 2019) and how does it gets learned (Marschall et al 2023), we see the birdcage as another opportunity to use a specific example dynamic system - task relationship as a way of making more general points  about the nature of learning across tasks.

---

> > ### Comment · Reviewer_MAk9 · 2025-11-25
> >
> > I thanks the authors for the clarifications; those address some of my concerns: (1) difference between this work and Driscoll et al. (2023) (2) discrete vs continuous time RNNs (3) intuition behind the emergence of a limit cycle during the task.

---

### Official Review · Reviewer_2N7E · 2025-11-02

**Soundness:** 3
**Presentation:** 4
**Contribution:** 3
**Rating:** 6
**Confidence:** 4

**Summary:**

This paper investigates how curriculum design affects transfer and representational geometry in recurrent neural networks (RNNs) trained on sequential sensorimotor tasks. The authors frame skill transfer as a dynamical-systems problem and analyze how pretraining on related subtasks alters the effective rank and phase geometry of population activity during subsequent learning.

They implement a family of structured curricula that vary task difficulty and temporal structure and compare them with single-task (“solo”) training on paradigmatic continuous-control tasks such as Integrate-n and Count-n.

The key findings are:

Pretraining on simpler or shorter tasks accelerates convergence on longer or more complex ones.

Transfer efficacy correlates with trajectory alignment and reduced effective rank in neural population activity.

Intermediate “mixed” curricula yield smoother representational transitions than direct or disjoint training.

Figures 7–8 visualize these effects through changes in effective rank and phase alignment of hidden-state trajectories.

**Strengths:**

1. The paper goes beyond behavioral speedups and analyzes the geometry of hidden trajectories, linking dimensionality reduction to efficient transfer — a clear and interpretable mechanistic contribution.
2. The chosen tasks (Integrate-n, Count-n) and architectures are simple enough to allow transparent interpretation; the analysis of effective rank and phase structure is well justified and reproducible.
3. Figures 7–8 clearly demonstrate how curriculum structure modulates trajectory alignment and representational compression, with consistent trends across curricula and seeds.
4. The work situates itself among both classical and modern studies of curriculum learning and RNN dynamics, citing Soviany et al. (2022), Schuessler et al. (2020), and Sussillo & Barak (2013).
5. Experimental details, data splits, and metrics are described thoroughly; the authors commit to releasing code upon publication.

**Weaknesses:**

Lack of formal theory.
There is no explicit model explaining why lower effective rank corresponds to faster transfer. The relationship is described qualitatively but not derived or predicted mathematically.

Narrow task family.
Experiments are confined to low-dimensional continuous-control tasks (Integrate/Count). It remains unclear whether the same representational mechanisms hold in higher-dimensional or non-periodic domains.

1. Omission of closely related theoretical work. The paper cites Proca et al. (2025) on learning dynamics but omits Rajan, Kepple, & Engleken ICLR, who analyzed sequential curricula and representational transfer in RNNs — a directly relevant body of work. This omission weakens the contextualization of the present results.

2. Limited quantitative rigor. While figures are compelling, the authors provide no statistical tests or uncertainty estimates for rank and phase measures; relying solely on visual consistency is insufficient for strong claims.

3. Weak link to cognitive and behavioral data. The discussion briefly mentions parallels to human skill-transfer and hierarchical learning, but these analogies are not developed, missing an opportunity to connect the findings to cognitive or neuroscience evidence. Summerfield and Saxe lab paper are useful refs.

**Questions:**

1. Can you formalize the observed relation between rank reduction and transfer speed—e.g., via a simple linear-subspace or covariance-alignment model?

2. Have you tested whether rank alignment persists if network recurrence weights are trained rather than frozen?

3. Could you provide confidence intervals or statistical tests for rank and phase-alignment metrics?

4. How would curricula behave under non-periodic tasks (e.g., context switching or delayed association)?

5. How does your mechanistic interpretation relate to the Rajan–Kepple–Engleken findings on representational reuse?

---

> ### Author Response · Authors · 2025-11-19
> **Reply**
>
> Weaknesses
>
> With regards to task family, we would argue that our family is not narrower or wider than the working memory variations from Yang used in the Driscoll work, the evidence integration / working memory in Rajan, Kepple, & Engleken or the Barak tasks. To expand the scope a little we’ve added flip flop task as a third pretraining option which helps us disambiguate effects of geometry vs topology.
>
> In terms of links to behavioral data: this is at the moment out of scope but something we would like to consider more closely going forward
>
> Questions:
> Q1) This is an interesting question, and there is existing work that explores natural rank reduction and training speed in the context of lazy and rich learning (Atanasov, Bordelon, Pehlevan. ICLR 2021), which we will briefly explain here. This work explores how finite-width kernel machines for teach-student architectures minimize mean-squared error losses from small initial parameter weights. They capture rich learning dynamics in RNNs by assuming that the kernel evolves in two phases:  i)  an alignment phase in which the kernel aligns to a low-rank structure, where the alignment time is set by the relative strength of the teaching signal, and ii) a refinement stage where the kernel only changes in scale. They find that this assumption agrees with empirical data, and that the first stage’s “silent alignment effect” explains how low-rank structure arises in data, and that the second stage can fine tune the kernel to perform the task.  Our work is in this training regime (MSE loss, small initial weights), and we believe that our dynamical systems results expands on this work by demonstrating a) how such low-rank structure manifests at the level of representation, and b) how only the second phase of fine tuning is needed for a second training tasks that shares this low-dimensional structure.  While out of scope for now, we’d argue that Atanasov et al  does provide some theoretical base for some the results, which we will highlight in the discussion.
>
> Q2) There may be a fundamental misunderstanding about our approach here. In our analysis, we always let the network train with full rank recurrent weights, we then use the low rank approximation of the final weights at test time, just to test how much of the functionality is still preserved after the low rank approximation.  This follows the standard procedure established in prior work analyzing learning-induced connectivity changes in RNNs (Schuessler et al., 2020). One could force the solution to be low rank during training and see how that behaves. Not clear what would that teach us though… Would you intuit this version to behave differently in a way that is diagnostic?
>
> Q3) We have replaced the original sem (too tiny to be visible) with CIs, and updated the figures and text accordingly. For the phase-alignment metric, we’re not sure what would be the null hypothesis to be tested against.
>
> Q4) We haven't tried context switching but we now include a simple item working memory version (2bit flip-flop task) which has similar low-dimensionality and the same number of slow points as a 4Count (minus the rotational flow in the latter). We could show that the former is in fact detrimental for learning (new appendix). Without the key dynamical system motif of the limit cycle, the low d very constrained structure hurts rather than helps new learning (similar in spirit to the effects of low rank initialization on RNN training in Liu et al 2025).
>
> Q5) When comparing to the Rajan–Kepple–Engleken results: the goals of the papers are different in that theirs looks for multi-task diagnostics of the underlying learning process whereas here we want to understand the mechanics of reusing slow motifs as combinatorial elements of neural computation. There is some similarity in task structure but even there we’d point out that our task requires reporting an online sum at each time point rather than the mean of the samples up to a confidence criterion. Finally, although some analyses of geometry seem potentially related, the focus here is on topology (axes of slowness, reuse and building new ones) which is unique to our work. Overall, we see the two as complementary but not strongly related. We will add some notes on the comparison in the discussion.

---

### Official Review · Reviewer_fgVg · 2025-11-04

**Soundness:** 2
**Presentation:** 2
**Contribution:** 2
**Rating:** 2
**Confidence:** 3

**Summary:**

The paper presents an empirical study on the structure of the dynamics of a simple RNN at different stages during training in the context of pre-training and curriculum learning for an “evidence integration” task. It uses various qualitative (e.g. attractors in the network activity) and quantitative measures (e.g. rank of weight change matrix) based on the hidden state trajectories, network activity, or weight matrix evolution to characterize the geometry of the network-dynamics and efficacy of different pre-training approaches (same task but simpler vs. different counting based task). They convincingly show differences in efficacy between the different pre-training / curriculum-training approaches, and subsequently compare them to similarities / differences in the geometry of the respectively trained network / network dynamics. Interestingly, the slightly different counting task turns out to be the more beneficial pre-training task.

**Strengths:**

1) The paper tackles interesting and relevant questions regarding the characteristics of good pretraining tasks, which are thoroughly motivated in its introduction.

2) The pretraining / curriculum learning setup seems novel and the chosen qualitative and quantitative metrics suitable for investigating the geometry of the resulting network / network dynamics of different pretraining approaches.

3) The paper clearly demonstrate differences in efficacy as well as geometric structure in the artifacts of the various pretraining approaches, and the geometric explanation of their efficacy seem like a promising research direction (e.g. reuse, reshape, and de novo formation).

**Weaknesses:**

W1) The abstract is difficult to understand; possibly due to convoluted writing and imprecise use of technical terms.

W2) The paper seems to lack a section on related work, making the contextualization of experimental setup, evaluation methods and results, as well as originality somewhat difficult.

W3) The results are shown for **one simple RNN** (Elman type) architecture and one type of integration problem which is the sum operation. No comparison of different architectures etc. Will you analysis also be applicable to LSTMs? As it is an empirical paper, I am not convinced about generality of the evidences and conclusions.

W4) The method section is mathematically imprecise (see mu sampling, possibly incorrect e.g. “continous time“ RNN even though equations show discrete time RNN) and seems to lack a lot of detail (e.g. how were the networks trained precisely, exact training setup, which libraries were used for computation, etc.). Also the appendix lacks these details. The paper is unlikely to be reproducible.

W5) Many figures feature cryptic labeling (e.g. figures 2-4), and could benefit from proper scaling and labeling of legends where missing, or additional details on abbreviations in their subtext.

W6) It is somewhat hard to reason about what exactly the experiments (and metrics) show in the end. It often doesn’t seem like an apples to apples comparison between the integration and counting pretraining task, as multiple dimensions are changed at once (e.g. periodicity AND target), and possibly some more informative ablations or controls are missing (e.g. Integrate 6A to Integrate 12A, or Initial to integrate 6A).

W7) The presented explanation for the efficacy of the pretraining in terms of geometry of the network / network-dynamics is not fully convincing / sufficiently substantiated. While differences in efficacy of the pretraining procedure are clear, and that differences in the geometric structure of the network / network dynamics exists due to the different pretraining also seems sufficiently evidenced; however, the argument / evidence as to how or why the pretraining induced geometry in network / network dynamics is better remains somewhat unclear. For example; how could I determine based on the induced geometry in network / network dynamics PRIOR to actually evaluating the final network training whether it will be more or less beneficial than a particular other geometry?

**Questions:**

Q1) How was the network trained precisely (loss function, optimizer, learning rate, epochs, batch size, etc.), and the details regarding the training data (exact sigma_stim, mu_0, and number of trials during training)

Q2) How common are networks that perform that task well but have no “bird cage“ structured (as mentioned in Lines 198-199 and Fig. 3D.) And what do their dynamical plots look like (analogous to 1C)?

Q3) For the analysis in terms of rank the network weight changes; was W_in, W_rec or W_out or all three analyzed? And if only W_rec why not all three?

Q4) What is the precise threshold mentioned in Figure 2C (L182-L183)

Q5) How were the perturbations sampled / done in Fig 1.C and 4.C?

Q6) What are the results for evaluating metrics in Figure 6B between e.g. Initial A to Count 6A, or Integrate 6A Integrate 12A? (This could serve as additional control.)

Q7a) How exactly does the geometry of the dynamics of a integration pretrained network look like; i.e. Figure 1C or 4B type of plot but for shorter integration T=6?. And is it more similar to Figure 1C than 4B?

Q7b) And if it is more similar, than doesn’t this conflict with the explanation that the low dimensional change to geometry of the dynamics is responsible for the quality of the pretraining? (At least 5B looks way more similar to 5E than 5A).

Q8) Do the results in Figure 3 “C -> Int” “final-initial” mean that the solution (weights) that are found by pretraining with counting are closer to initializaiton (weights) than they would be by using no pretraining or short integration?

Q9) What would the comparison in figure 2 look like, if counting was also pretrained with “short“ sequences? From the graphic it might seem that simply pretraining with any task that has matching cyclicity is the core importance. Not the counting itself.

Q10) How does the integration pretrainig task “hone sub-skills“ more so than using the same task but with shorter sequences? And is this also true if integration pretraining was also half periodicity?

Q11) The question from weakness 7).

Q12) How exactly do the results support “cognitive flexibility“?

---

> ### Author Response · Authors · 2025-11-19
> **Response to reviewer weaknesses**
>
> W1-3:
> Regarding the general style of the writing: the format is common for papers within the computational neuroscience segment of the community, it is not clear why it should be evaluated by machine learning priors of structure. The ICLR format does not require a related work section and the content of the comparison is in fact provided as part of the Introduction and the discussion.
>
>  Regarding the choice of RNN architecture, again since the goal is to understand knowledge transfer at the level of reuse of dynamics in brain circuits vanilla RNNs are the only relevant architecture, and this is completely aligned with all past relevant literature that uses either vanilla RNNs or even simpler linear RNNs for proofs. In terms of the range of tasks considered: the closest related work from Barak, Driscoll, Rajan, etc all restrict their analysis to a small number of tasks of neuroscience relevant. So we would argue that we are not outside of the norm for the domain in that aspect either.
>
> W4: We have updated the text for more methods details; we are planning to provide the full code with the paper which should guarantee reproducibility. There is no reason to assume that any of the hyperparameter choices which were only briefly  described in the original submission have a qualitative impact on any of the results presented.
>
> W 5-6: There are some fundamental misunderstandings about the experiments which explain the confusion about how the comparison to controls was done and why which we hope to clarify in the detailed reply to questions. The comparison is apples to apples in the sense that we’re matching, architecture, learning details, convergence criteria etc in networks with matched seeds, leaving the only dimension of variation the nature of the pertaining task and what kind of dynamics it induced in the RNN.
> W7: We have also added a new set of results that further disambiguate the effects of geometry (small but present) from topology (the main effect and the predictor of successful transfer)

---

> ### Author Response · Authors · 2025-11-19
> **Detailed answer to questions 1-12**
>
> R1
>
> Q1: We apologize for any unclarity, the optimization used Adam on an mse loss, with a learning rate of 0.0001. The batch size was 32, and 512 batches were processed per epoch. Network convergence was defined by the test MSE being below 0.2 for 10 consecutive epochs. details in appendix.
>
> Q2) As stated (L198-99), alternative solutions are reasonably common. For Integrate6 11 out of 25 seeds performed the task ok but did not exhibit the clear "birdcage" structure. The dynamics for these are typically high-d, making their dynamical structure difficult to interpret intuitively.  Implicit regularization by longer training might reduce their occurrence. Importantly, pretraining helps guide the networks robustly to the low-d solution, eliminating this variability.  100% of seeds pretrained on counting converge to the birdcage.
>
> Q3) The minimal rank is a property of recurrent weights. The rank of the input and output weights is already low, as determined by task structure.
>
> Q4) The convergence threshold is an MSE of 0.2, used for all training stages across all experiments. The results are qualitatively robust and do not strongly depend on this specific choice.
>
> Q5) The perturbation methods for 1E and 4C were different. For Figure 1E, the perturbations were isotropic Gaussian noise. The autonomous (zero-input) dynamics were then plotted from these perturbed initial conditions. For Figure 4C, the perturbations were structured and directional to specifically test the attractor properties orthogonal to the counting limit cycle. This was found by running PCA on autonomous trajectories originating from states with small initial noise, and then selecting the third principal component.
>
> Q6) We’ve added a mean estimate  for InitialA-CountA as baseline in Figure 6B. The network states for integrate6 vs the final solution of integrate6 → integrate12 were far more similar to each other (across PC-axis similarity, subspace alignment, and CKA) than either was to the solo integrate12 network.
>
> Q7) If we correctly understand the first question, this was already shown in fig 5B. Integration 6 will have a 6 bar birdcage structure and need to add 6 more during the integrate 12 training, how the extra elements get added is not completely stereotyped, sometimes the final solution looks more similar to the integrate 12 trained from scratch, sometimes the solution looks more like integrate 6 with a double cycle type of structure (for int/count multiples)
>
> How the difference in the number of bars final vs pretraining relates to the rank change is not trivial. But we also would not want to put too much emphasis on the rank as such.   New results show that this is down to topology not just geometry (see appendix).
>
> Q8) I suppose it is a matter of what metric one uses to assess closeness but not in any obvious way. Also it is worth remembering that we measure the smallest low rank approximation of the final (full rank trained solution) that can still satisfy a pre-set performance criterion, so small rank means that the final function is concentrated into a low d space not necessarily that there are no changes in the other dimensions.
>
> Q9) We have shown that it is not critical that the count number is matched to the target trial length. It is a bit of a subtle point but Fig2 includes 3 different experiments  for each target. For example, the target integration length of 12 (shown in the same green color) includes three separate data points and error bars for Count 6,8,12 pretraining.  The results strongly overlap for these, showing that specific trial length is not importan; the limit cycle build by counting is.
>
> Q10) A simple way to think about it is that speed comes from the limit cycle. Short integration can be hit or miss in terms of building that structure, while count is 100% reliable. The target being multiples of the short or not does not affect this mechanism. When a smaller set of bars is reliably produced (e.g. via count6 followed by integrate6) that extra structure can be reliably reused as well.
>
> Q11) Completely unequivocal evidence would require formal proofs, which we do not have at the moment. Nonetheless, we do now include additional results trying to disambiguate the contribution of geometry vs topology, which argue that the details of the dynamics matter just as much or more as the rank of the pretraining and target  task solution.
>
> Q12) The idea is that (through past experience or over development) the brain constructs a bank of such useful dynamical systems motifs which it can then use and reshape in the service of new tasks and learning. This point has been made in the past fixed dynamics motifs where the flexibility comes from learning new inputs into and outputs out of the network (e.g. Driscoll ). The novel aspect here is that the motifs themselves get to be reshaped as they adapt to new uses, while still preserving their original function.

---

> > ### Comment · Reviewer_fgVg · 2025-11-26
> > **Response to authors.**
> >
> > Thank you for taking the time to address the critique and questions, and updating the manuscript.
> > Answering the above questions, adding details on the methods in the appendix (and committing to releasing the experiment code), and adding further disambiguating experiments helps to improve the soundness of the experiments and results.
> >
> > Nonetheless, significant doubts wrt. the soundness of the results remains esp. in the direction of previous critique W3, W6 and W7. In particular also keeping the training methods the same between two ablations, but nonetheless changing the training dataset which qualitatively differs along TWO dimensions (e.g. periodicity & target) makes it hard to isolate / attribute the observed effect.
> >
> > Similar conceptual problems applies to other experiments (hence missing ablations / baselines).
> > Furthermore, the paper could still benefit from an improved abstract, a more clear positioning and disambiguation to related literature (e.g. in line with Reviewer 2N7E and MAk9), better legends, and more precise experiment / ablation descriptions etc. This critique is NOT about wanting a specific “ICLR style“ of writing. This is important for readers not already deeply familiar with the related literature, and would increase readability and accessibility for a wider audience.

---

### Meta-Review · Area_Chair_deAc · 2025-12-24

**Summary:**

This paper qualitatively investigates the potential underlying mechanisms of skill transfer from pretraining to target tasks in RNN, from a dynamical system point of view. Major concerns from the reviewers are

- Limited novelty and scope
- Lack of formal theory
- Task generalizability
- Network architecture generalizablity
- Weak link to cognitive and behavioral data
- Writing and presentation

Meanwhile, the reviewers also acknowledged the contributions of this study, such as

-  Tackling important and well-motivated questions
-  A clear and interpretable mechanistic contribution
-  The geometric explanation seem like a promising research direction

Overall, I believe this paper brings out an interesting mechanistic investigation on a timely and significant problem in computational neuroscience and deep learning, which, while arguably had been discussed by previous studies, is certainly valuable and has its unique contribution. Meanwhile, many concerns discussed by the reviewers also make sense and are worthy directions to improve the paper. Therefore, I would encourage the authors to perform a major revision according to the review comments---this is not to say the current manuscript does not make a significant contribution to the community---but rather, incorporating the changes (imroving writing and presentation, testing on more complex tasks and networks, etc.) will escalate this work to impact boarder audience.

**Reviewer Concerns:**

The authors have answered most of the questions by the three reviewers. However, several key concerns still remain

1. **lack of for rigorous quantitative justification**(fgVg, 2N7E): The authors acknowledged they offer a qualitative mechanistic explanation rather than a formal derivation, leaving this theoretical gap unfilled
2. **Weak link to cognitive and behavioral data** (2N7E): The authors left this as future work.
3. **Unclear generalizability** (all reviewers): The authors defended their scope by citing field norms, but the concern that these mechanisms might not translate to broader ML contexts remains.

**Reviewer Scores:**

As discussed above, key concerns remain. I would not expect a significant up-rating.

---

### Decision · Program_Chairs · 2026-01-26

Reject